# In Vitro and In Vivo Therapeutic Potential of 6,6′-Dihydroxythiobinupharidine (DTBN) from *Nuphar lutea* on Cells and K18-*hACE2* Mice Infected with SARS-CoV-2

**DOI:** 10.3390/ijms24098327

**Published:** 2023-05-05

**Authors:** Shay Weiss, Kamran Waidha, Saravanakumar Rajendran, Daniel Benharroch, Jannat Khalilia, Haim Levy, Elad Bar-David, Avi Golan-Goldhirsh, Jacob Gopas, Amir Ben-Shmuel

**Affiliations:** 1Department of Infectious Diseases, Israel Institute for Biological Research, Ness Ziona 7410001, Israel; shayw@iibr.gov.il (S.W.); eladb@iibr.gov.il (E.B.-D.); amirb@iibr.gov.il (A.B.-S.); 2The Shraga Segal Department of Microbiology, Immunology & Genetics, Faculty of Health Sciences, Ben-Gurion University of the Negev, Beer Sheva 8410501, Israel; kamranwaidha1@gmail.com (K.W.);; 3Chemistry Division, SAS, Vellore Institute of Technology, Chennai Campus, Chennai 600127, India; 4Department of Pathology, Soroka University Medical Center and Faculty of Health Sciences, Ben-Gurion University of the Negev, Beer Sheva 8410501, Israel; 5The Jacob Blaustein Institutes for Desert Research (BIDR), Ben-Gurion University of the Negev, Sede Boqer Campus, Sde Boker 8410501, Israel

**Keywords:** SARS-CoV-2, COVID-19, 6,6′-dihydroxythiobinupharidine (DTBN), RNA-dependent RNA polymerase (RdRp), anti-viral small molecule drug

## Abstract

We have previously published research on the anti-viral properties of an alkaloid mixture extracted from *Nuphar lutea*, the major components of the partially purified mixture found by NMR analysis. These are mostly dimeric sesquiterpene thioalkaloids called thiobinupharidines and thiobinuphlutidines against the negative strand RNA measles virus (MV). We have previously reported that this extract inhibits the MV as well as its ability to downregulate several MV proteins in persistently MV-infected cells, especially the P (phospho)-protein. Based on our observation that the *Nuphar* extract is effective in vitro against the MV, and the immediate need that the coronavirus disease 2019 (COVID-19) pandemic created, we tested here the ability of 6,6′-dihydroxythiobinupharidine DTBN, an active small molecule, isolated from the *Nuphar lutea* extract, on COVID-19. As shown here, DTBN effectively inhibits SARS-CoV-2 production in Vero E6 cells at non-cytotoxic concentrations. The short-term daily administration of DTBN to infected mice delayed the occurrence of severe clinical outcomes, lowered virus levels in the lungs and improved survival with minimal changes in lung histology. The viral load on lungs was significantly reduced in the treated mice. DTBN is a pleiotropic small molecule with multiple targets. Its anti-inflammatory properties affect a variety of pathogens including SARS-CoV-2 as shown here. Its activity appears to target both pathogen specific (as suggested by docking analysis) as well as cellular proteins, such as NF-κB, PKCs, cathepsins and topoisomerase 2, that we have previously identified in our work. Thus, this combined double action of virus inhibition and anti-inflammatory activity may enhance the overall effectivity of DTBN. The promising results from this proof-of-concept in vitro and in vivo preclinical study should encourage future studies to optimize the use of DTBN and/or its molecular derivatives against this and other related viruses.

## 1. Introduction

Safe and therapeutically useful antiviral compounds are needed to confront the emergence of new viruses, combat viral infections and prevent drug resistance. Natural products from a variety of sources, some based on ethno-medicine traditions, are an important source for antiviral drug development. The possible targets for these new drugs may be of viral or cellular origins, inhibiting viral internalization, replication or release.

We have previously published research on the ability of an alkaloid semi-purified mixture extracted from the aquatic plant *Nuphar lutea*, which consists mainly of dimeric sesquiterpene thioalkaloids called thiobinupharidines and thiobinuphlutidines [1], to inhibit the negative strand RNA measles virus (MV) [2]. We have previously reported that this extract inhibits the MV as well as its ability to downregulate several MV proteins in persistently MV-infected cells, especially the P (phospho)-protein. 

The RNA-dependent RNA polymerase (RdRp) replication of negative-strand RNA viruses relies on two components: a helical ribonucleocapsid and an RNA-dependent RNA polymerase composed of a catalytic subunit, the L protein, and a cofactor, the P protein. 

Based on our observation that the *Nuphar* extract is effective in vitro against the MV, and the immediate need that the COVID-19 pandemic created, which in turn spurred the pharmaceutical industrial sector as well as academia to run parallel programs for the development of vaccines, monoclonal antibodies, peptides and small molecules for the management of the virus, we tested the ability of 6,6′-dihydroxythiobinupharidine DTBN, an active small molecule, isolated from the *Nuphar lutea* extract, on COVID-19, Figure 1.

Several recent thorough reviews have been published [3,4,5,6] on the state-of-the-art status of small and peptide-based molecules in the detection of the virus and interaction with the microbiome [7], as well as therapeutic monoclonal antibodies against COVID-19.

COVID-19 is an RNA-positive single-strand novel beta-corona virus which is the cause of the severe acute respiratory syndrome, SARS-CoV-2. The viral infection is highly transmissible and starts by the entry of the virus through its cellular receptor followed by endocytosis and fusion of the membrane. The interaction of the receptor-binding domain (RBD) of the viral spike glycoprotein with the angiotensin-converting enzyme 2 (ACE2) enables its internalization. Viral replication and the release of virions with infective capacity follows. 

Vaccination and treatment are the two essential arms that limit the epidemic and its cure. Several monoclonal antibodies, peptides and small molecule inhibitory drugs are being tested and used, complementing each other. 

A variety of plausible molecules are being targeted, specifically the 3C-like protease (3CLpro), papain-like protease (PLpro) and non-structural viral proteins (NSPs) important in virus replication. Nirmatrelvir, a protease inhibitor, has been approved. RNA-dependent RNA polymerase (RdRp) and helicase inhibitors have been also approved by the Food and Drug Administration such as remdesivir and molnupiravir [8,9,10].

RdRp (nsp12) represents a well-explored target for the design of antiviral inhibitors. As such, RdRp catalyzes the viral genome synthesis, which serves an essential role in coronavirus replication. Both nucleosides as well as non-nucleoside-based drugs have been explored as RdRp inhibitors; however, the development of non-nucleoside-based inhibitors has often encountered hurdles owing to the issue of drug resistance.

The rapid spread of infection by SARS-CoV-2 and its variants fully warrants the continued evaluation of drug treatments for COVID-19. Here, we explored the therapeutic potential of 6,6′-dihydroxythiobinupharidine (DTBN) purified from the *Nuphar lutea* extract, in vitro on SARS-CoV-2-infected Vero E6 cells, and in vivo on SARS-CoV-2-infected K18-*hACE2* mice expressing the human ACE2 receptor.

The results from this proof-of-concept preclinical study encourage us to further investigate this compound against this and other viruses.

## 2. Results

### 2.1. Determination of Half Maximal Inhibitory Concentration (IC50) of DTBN on Vero E6 Cells

To test the antiviral activity of the DTBN inhibitor against SARS-CoV-2, the dose used to inhibit 50% of SARS-CoV-2 was calculated for DTBN by determining the inhibitory concentration 50 (IC50), as shown in Figure 2A. Vero E6 cells were incubated with serial dilutions of DTBN 1 h before infection. Then, Vero E6 cells were infected with SARS-CoV-2 at a multiplicity of infection [MOI] = 0.01. At 72 h post infection cell viability was measured using the cell proliferation assay (XTT based). The IC50 value of DTBN (IC50 = 0.29 μM) was determined at a low micromolar concentration. To determine the DTBN cytotoxicity, Vero E6 cells were incubated with serial dilution of DTBN for 72 h and cell viability was determined (Figure 2). For comparison, a positive control experiment with the same cells and conditions was carried out with remedesivir. The IC50 of remedesivir is effective at lower concentrations and less toxic than that of DTBN.

### 2.2. DTBN Pretreatment of Vero E6 Cell Prior to Infection, Inhibits the Release of Infective Viral Particles to the Supernatant

Vero E6 cells were incubated with 1 μM DTBN for 1 h and then infected with SARS-CoV-2 MOI-0.01. Supernatants were harvested 24 h after infection and analyzed by plaque-forming units (PFU) assay to measure the effect of the DTBN on SARS-CoV-2 replication (Figure 3). Approximately 7 × 10^6^ PFU/mL were detected in the medium of vehicle dimethyl sulfoxide (DMSO) (untreated) infected cells, whereas no plaques were detected in DTBN-treated cells, indicating a significant inhibition of the virus release.

### 2.3. DTBN Treatment of Vero E6 Cell following Infection, Inhibits the Release of Infective Viral Particles to the Supernatant

Vero E6 cells were infected with SARS-CoV-2 MOI-0.01, 24 h post-infection the cells were incubated with 1 μM DTBN. Supernatants were harvested 48 h after infection and analyzed by plaque-forming units (PFU) assay to measure the effect of the DTBN on SARS-CoV-2 replication (Figure 4). Approximately 3E5 PFU/mL were detected in the medium of vehicle dimethyl sulfoxide (DMSO) (untreated) infected cells, whereas no plaques were detected in DTBN treated cells.

### 2.4. Survival of Infected Mice Treated with Vehicle or with DTBN

In order to determine whether the inhibition of the virus in vitro translated into a therapeutic benefit in mice, we evaluated the survival of mice treated with vehicle or with DTBN. As a proof of concept, we chose a short-term protocol where the animals were virally infected and treated concomitantly for 5 days. The mice were further monitored for 14 days. The results in Figure 5 show a significant curative benefit of DTBN treated mice; 30% (4/12) of the DTBN treated mice survived. 

### 2.5. Expression of Viral Particles in Lungs of Vehicle and DTBN Treated Mice

Paraffin blocks and sections of lungs were prepared 3 days after viral infection and concomitant treatment with vehicle or DTBN. A total of four lungs of vehicle-treated mice and three lungs from DTBN-treated mice were stained, scanned and analyzed independently. 

Sections from vehicle or DTBN treatments were first H&E stained to evaluate the tissue’s histology. Sections of the same lungs were stained for virus presence and representative sections are shown in Figure 6. H&E staining of lungs from virally infected but vehicle-treated mice showed cluttered airways with atypic cells, without clear consolidation (Figure 6A). DTBN-treated mice showed normal histology with slight emphysema and some congestion of blood vessels (Figure 6B). Sections of control uninfected mice, treated with vehicle or with DTBN, presented a normal histology and an absence of viral particles (Appendix A). 

To evaluate the presence of virus in the lungs, fluorescence immunohistochemistry was performed with a polyclonal anti-COVID-19 antibody. The ratio of total tissue area to total virus area was calculated for each image (four lungs of vehicle-treated mice and three lungs from DTBN-treated mice). The unpaired *t*-test of the area ratios was computed, showing a significant difference between the control and two treated groups (*p* value = 0.0215). For detailed data see the Appendix A. Sections of representative lungs, one from vehicle-treated (Figure 6C) and the other from DTBN-treated mice (Figure 6D), are presented. An example of normal (untreated and uninfected) H&E-stained lung tissue is shown in the Appendix A.

The results show a significant reduction in virus presence in the DTBN-treated group (Figure 6E) which correlates with the better survival of the DTBN-treated mice. 

### 2.6. In Silico Molecular Docking of DTBN and suramin to SARS-CoV-2 RNA Dependent RNA Polymerase (RdRp)

In the present study DTBN shows promising inhibitory activity against SARS-CoV-2 in in vitro and in vivo studies. Since docking studies determined that DTBN was inactive against the molecular target, M^pro^ of SARS-CoV-2 [11], we undertook a docking study to improve the understanding of protein-ligand, sites and types of interactions existing between ligand (DTBN) and other possible target proteins, RdRp. DTBN docked to the catalytic site of RdRp and ACE2. To validate the docking protocol, suramin bound to the crystal structure was extracted and re-docked to its crystal structure. Furthermore, superimposed over its native suramin-bound crystal structure, the RMSD values for c- alpha atoms and ligand were found to be 0.18 Å and 1.4 Å respectively, thus validating the docking protocol [12]. The superimposition and interaction of native and docked is given in Appendix A. The RdRp domain of SARS-CoV-2 comprise 7 motifs ranging from 499 to 820. A detailed description of the motifs is given in Table 1.

DTBN resides at the cavity formed by conserved motifs G and N terminus of Motif B of RdRp. At the active site, one of the hydroxy hydrogens of DTBN is involved in the hydrogen bond with phenolic oxygen of Tyr689 ((O)H⋯O(H), 2.62 Å). The other hydroxy group of DTBN engages in three hydrogen bonds: (i) hydroxy hydrogen of DTBN involved in hydrogen bond with amide carbonyl oxygen of Asn497 ((O)H⋯O(C), 2.16 Å), (ii) hydroxy oxygen of DTBN involved in two hydrogen bonds, one with guanidinium hydrogen of Arg569 ((H)O⋯H(HN^+^), 2.32 Å) and the third is with ammonium hydrogen of Lys500 ((H)O⋯H(H_2_N^+^), 2.39 Å). In addition to the above hydrogen bond interactions, DTBN is stabilized by aromatic C-H⋯O hydrogen bonds: C-H of furan ring of DTBN involved in hydrogen bond with amide carbonyl of Asp684. Furthermore, the ligand–protein interaction was reinforced by a pi–cation interaction (pi electrons of furan ring of DTBN interacts with –NH_3_^+^ of Lys577). 

Apart from its stabilizing interaction, DTBN experiences two steric clashes at the catalytic site. A steric clash is observed between the quinolizine ring of DTBN and the amino acid residue of Lys577 and Tyr689 (Figure 7).

To scale the reactivity of DTBN, we compared the binding energy with a potent RdRp inhibitor, suramin and MLN-4760, an ACE2 inhibitor (Table 2). The binding energy metrics show that suramin has a higher affinity in comparison to DTBN. High binding energy and affinity are associated with a wide array of hydrogen bonds. At the active site, suramin is stabilized by eight hydrogen bonds, one aromatic C-H⋯O hydrogen bond, one pi–cation and salt bridge interactions. Hydrogen bonds are observed between amino acid residues, Asn496, Asn497, Lys500, Arg569, Gln573, Lys577, Thr591 of RdRp and -NH_2_, -SO_3_H and -CONH_2_ of suramin. Pi–cation interaction exists between Lys577 and the naphthalene ring of suramin. The salt bridge is observed between amino acid residue, Arg569 of RdRp and the sulfonyl oxygen of suramin. Whereas MLN-4760, at the active site of ACE2, was found to be stabilized by eight Hydrogen bond interactions with amino acid residues: Arg273, His345, Pro346, Glh375, His378, His505 and Tyr515. Additionally, MLN-4760 was found to be stabilized by a pi–cation interaction with His374.

To further understand the possible mechanism of RdRp inhibition by DTBN, the RdRp–DTBN complex was further compared with the approved drug remdesivir which binds and inhibits the RdRp complex (PDB id: 7BV2). We noticed that at the catalytic site DTBN occupies the space of the RNA template strand suggesting that DTBN may not only disrupt the binding of the RNA template–primer duplex but may also prevent the entry of nucleotide triphosphate into the catalytic site, resulting in the direct inhibition of RdRp activity (Figure 8). The ΔG for remdesivir docked was found to be −51.81 Kcal/mol which is corroborated by the experimental results showing that suramin is more potent than remdesivir [13].

At the active site of ACE2, DTBN was found to be stabilized by two hydrogen bonds: (i) one of the hydroxy hydrogens of DTBN is involved in hydrogen bonding with the carboxylic oxygen of Ala 348 ((O)H⋯O(C), 1.91 Å) and the other (ii) between the furan ring oxygen of DTBN and imidazole -HN of His345 ((O⋯H(N), 2.14 Å). Additionally, DTBN is stabilized by two pi–pi interactions between the furan ring of DTBN and both (i) His345 and (ii) Tyr510. In addition, pi–cation interaction is found between the furan ring and Arg514. 

### 2.7. Molecular Dynamic (MD) Simulation 

In addition to the molecular docking study on the basis of binding scores, MD simulation of RdRp was conducted to determine the stability of the protein–ligand complex, the stability of protein, as well as to find the persistent protein–ligand interactions that were observed during the static molecular docking study. The MD simulation study complements the experimental data. The stability of the protein–ligand complex and protein can be traced by root-mean-square standard deviation (RMSD) trajectory. 

The protein–ligand complex was subjected to 100 ns MD simulation and the RMSD graph was followed to determine the stability of protein–ligand complexes (RdRp–DTBN and RdRp–suramin) and protein (RdRp). It was observed that deviation in the RMSD trajectory of protein in both the protein–ligand complexes (RdRp–DTBN and RdRp–suramin) is well within 4 Å suggesting that protein is stable when in complex with DTBN/suramin. The corresponding 2D picture is portrayed in Figure 9a,b. The results suggest that protein–ligand complexes should be stable under both in vitro and in vivo conditions. During MD simulation, three stable and persistent hydrogen bonds with more than 50% strength are observed in the RdRp–suramin complex, while a single persistent hydrogen bond with more than 50% strength was observed in the RdRp–DTBN complex. Mainly the hydroxy group of DTBN is involved in hydrogen bonding with amino acid residue Tyr689 of RdRp (Figure 10). In suramin, (i) sulfonate oxygen forms a hydrogen bond with amino acid residue Arg569, (ii) Amide carbonyl oxygen forms a hydrogen bond with Gly 590 and (iii) Amine –NH forms a hydrogen bond with Thr 591 of RdRp (Figure 11). 

The molecular interactions existing during MD Simulations were continuously monitored and captured in the contact histogram bar graphs (Appendix A). The interaction of DTBN and suramin with specific amino acid residues of RdRp during MD simulation is plotted as a bar graph between interaction fraction vs. amino acid residue. These interactions were categorized into four types: hydrogen bond, hydrophobic, ionic and water bridges. The contact histogram bar graph clearly shows that RdRp interacts better with suramin than with DTBN, and this explains why the suramin–RdRp complex has a higher binding energy than the DTBN–RdRp complex. Yet, this analysis suggests that DTBN may exhibit a potentially inhibitory activity against SARS-CoV-2, possibly by inhibiting at least one viral target, RdRp. Thus, DTBN, a natural product, has the potential to be further developed as a potent antiviral drug. 

## 3. Discussion

Since SARS-CoV-2 replication leads to many of the clinical manifestations of COVID-19, antiviral therapies are being investigated for the treatment of COVID-19. These drugs prevent viral replication through various mechanisms, including blocking SARS-CoV-2 entry, inhibiting the activity of SARS-CoV-2 3-chymotrypsin-like protease (3CLpro) and RNA-dependent RNA polymerase (RdRp), and causing lethal viral mutagenesis [8] that may have the greatest impact before the illness progresses to the hyperinflammatory state that can characterize the later stages of disease, including critical illness. For this reason, it is necessary to understand the role of antiviral medications in treating mild, moderate, severe, and critical illness in order to optimize treatment for people with COVID-19.

One important obstacle in most studies searching for anti-viral agents is their cytotoxicity. Therefore, we determined the range of DTBN concentrations which were not cytotoxic to Vero E6 cells and in treated mice. Non-cytotoxic concentrations were determined and used to further investigate its anti-viral effect. 

DTBN and the extract from *Nuphar lutea*, are pleiotropic in their action. Regarding the semi-purified extract, it has anti-inflammatory activity by downregulating NF-κB [1,14] and partially preventing LPS-induced septic shock and peritonitis [14]. It is also effective against free as well as intracellular *Leishmania major* parasites [15,16,17]. The extract has anti-metastatic properties, synergistically with conventional chemotherapy drugs, both in vitro and in vivo. It also induces phospho-extracellular signal-regulated kinases (ERK) expression [18]. 

We have recently published research showing that the extract-purified DTBN molecule primes neutrophils against bacteria present in gum inflammation, enhances phagocytosis, reactive oxygen species (ROS) production, and NET formation [19]. We also reported on the in vitro antileukemic effects of DTBN by induction of apoptosis, correlated with significant biphasic changes in ROS cytosolic levels, a decrease in early time points (2–4 h) and an increase after a longer incubation (24 h) [20]. DTBN very efficiently and covalently inhibits human type II topoisomerase [21] conventional Protein Kinase Cs (PKC), most efficiently PKC alpha and PKC gamma [22], as well as inhibiting cysteine proteases such as cathepsins S, B and L and papain [11]. Interestingly, it does not inhibit M^pro^ of SARS-CoV-2. The inability to inhibit the viral protease suggests that DTBN acts on a different molecular target and not by inhibiting this viral protease [11]. 

Although from our earlier studies we know that DTBN cannot inhibit M^pro^, we show in the current study that DTBN inhibits the SAR-COV-2 infection, thus other targets must be available. Docking studies suggest that DTBN is not able to effectively bind to ACE2, suggesting that the DTBN anti-viral effect is due to its binding to a different viral and/or cellular protein. Since the plant extract very efficiently inhibits the measles virus P- protein which complexes to its RdRp, we analyzed the docking ability of DTBN within the catalytic domain of the SAR-COV-2 RdRdP, disrupting the binding to the RNA template–primer duplex and also preventing the entry of nucleotide triphosphates into the catalytic site. Its inhibition efficiency is further enhanced due to DTBN’s potential to inhibit other host proteins such as PKCα, Cathepsin L and NF-κB, which also play crucial roles in viral infection and replication.

NF-κB upregulation is observed in SARS-CoV-2 infection leading to the induction of uncontrolled inflammation [23,24], and it is observed that the spike receptor-binding domain or S-RBD and its binding to integrin α5β1 can cause NF-κB’s nuclear translocation by IκBα degradation.

They reported that the spike protein of SARS-CoV-2 alone can activate the NF-κB signaling pathway in a manner dependent on integrin ⍺5β1 signaling. The RGD domain with the RBD domain binds to integrin ⍺5β1 activating NF-κB target gene expression [24], and this study is corroborated by others [5,25]. These studies suggest that the induction of the NF-κB signaling pathway, and the subsequent cytokine production in endothelial (lung) cells, may be responsible for the cytokine storm that was observed in acute cases of SARS-CoV-2. The ability of DTBN to inhibit NF-κB, inhibit pro-inflammatory and induce anti-inflammatory cytokines is consistent with its anti-viral effect in vivo. Similarly, it has been suspected that cellular kinases such as PKCs have been involved in viral infection regulation of the influenza virus, HIV-1, and hepatitis E virus. SARS-CoV-2 decreases the activity and levels of human epithelial sodium (Na(+)) channels (ENaC) as regulated by PKCα/β1 and PKCζ activation [26]. Huang et al. reported on the impairment of replication SARS-CoV-2 by PKC inhibitors such as enzastaurin, Go 6983, sotrastaurin, rottlerin, and bisindolylmaleimide by the blocking of the viral entry and protein translation [27]. Cathepsin L (CTSL) plays an important role in COVID-19 infection and is elevated in patients infected with SARS-CoV-2 [28]. CTSL consists of an L domain α-helix and an R domain of β-sheet in its structure and is a lysosomal cysteine protease. It plays a major role in the proteolysis of antigen proteins that are produced via endocytosis by pathogens [29]. Previous in vitro studies have reported that SARS-CoV-1 depends on the proteolysis of its S protein by host CTSL [30]. The cysteine proteases CTSL and Cathepsin B, moderates cleavage of the Spike protein of SARS-CoV-1, which is required for the virus to gain entry into the host cell [31]. Zhou, P. et al. (2020) reported that SARS-CoV-2 shares 79.6% of its sequence identity with SARS-CoV-1 [32], and it may be inferred that SARS-CoV-2 also uses a similar path to enter host cells via CTSL. The study from Hashimoto et al. states that type II transmembrane serine proteases and cathepsin B and L contribute to the cleavage of Spike proteins and are essential for SARS-CoV-2 to infect cells [33], therefore making cathepsin L and B a promising target against COVID-19 infection. 

Topoisomerase 1 (TOP1) inhibition suppresses lethal inflammation induced by SARS-CoV-2.

Therapeutic treatment with two doses of topotecan (TPT), an FDA-approved TOP1 inhibitor, suppresses infection-induced inflammation in hamsters. TPT treatment as late as 4 days post-infection reduces morbidity and rescues mortality in a transgenic mouse model. These results support the potential of TOP1 inhibition as an effective host-directed therapy against severe SARS-CoV-2 infection [34]. In addition, *Mitoxantrone*, a type II topoisomerase inhibitor, has been reported to effectively inhibit the entry of spike pseudotyped viral particles and authentic SARS-CoV-2 virions by targeting both DNA topoisomerases and heparan sulphate in mammalian cells [35].

As shown here, DTBN effectively inhibits SARS-CoV-2 production in Vero E6 cells at non-cytotoxic concentrations. The short-term daily administration of DTBN to infected mice delays the occurrence of severe clinical outcome, lowers virus levels in the lungs and improves survival with minimal changes in lung histology. The viral load on lungs was significantly reduced in treated mice.

DTBN is a pleiotropic small molecule with multiple targets. Its anti-inflammatory properties affect a variety of pathogens including SARS-CoV-2 as shown here. Its activity appears to target both pathogen-specific as well as cellular proteins, such as NF-κB, PKCs, cathepsins and topoisomerase 2, that we have previously identified in our work. Although DTBN seems to be more limited in its antiviral effect in vitro as compared to remdesivir, the addition of DTBN to the therapeutic arsenal against the virus may offer an important advantage since DTBN seems to target both pathogen molecules, in our case SARS-CoV-2, as well as cellular components reported by others to be important in the ability of the virus to infect cells, replicate and induce a devastating pro-inflammatory storm. Thus, this combined double action of virus inhibition and anti-inflammatory activity may enhance the overall effectivity of DTBN. The promising results from this proof-of-concept in vitro and in vivo preclinical study should encourage future studies to optimize the use of DTBN and/or its molecular derivatives against this and other related viruses.

## 4. Materials and Methods

### 4.1. Cells

Vero E6 cells (ATCC^®^ CRL-1586TM) were maintained in Dulbecco’s Modified Eagle’s Medium (DMEM) supplemented with 10% fetal bovine serum (FBS), MEM nonessential amino acids, 2 nM L-glutamine, 100 units/mL penicillin, 0.1 mg/mL streptomycin and 12.5 units/mL nystatin (Biological Industries, Kibbutz Beit Haemek, Kibbutz Beit Haemek, Israel). The cells were cultured at 37 °C in a 5% CO_2_ and 95% air atmosphere.

### 4.2. Virus

The SARS-CoV-2 isolate Human 2019-nCoV ex China strain BavPat1/2020 was kindly provided by Prof. Dr. Christian Drosten (Charité, Berlin, Germany) through the European Virus Archive–Global (EVAg Ref-SKU: 026 V-03883). The viral stocks were propagated (4 passages) and titrated in Vero E6 cells.

### 4.3. Cell Viability

Cell viability was determined by XTT the Cell Proliferation Kit (XTT based) (Biological Industries Israel, 20-300-1000l) according to the manufacturer’s protocol. Vero E6 cells were seeded at a density of 3 × 10^4^ cells per well in 96 well plates or 4 × 10^5^ in 12 well plates (PFU assay). 

After overnight incubation, cells were treated in 3 replicates with 6,6′-dihydroxythiobinupharidine (DTBN) (Figure 1). DTBN was purchased from Sigma/Merck (SMB00609).

Cells were infected 1 h later with SARS-CoV-2 (MOI, 0.01). For PFU quantification, the supernatant was collected 24 h after infection. Cell viability was determined 72 h after infection by the XTT assay. For PFU quantification, Vero E6 cells were seeded at a density of 4 × 10^5^ cells per well in 12 well plates. After overnight incubation, cell monolayers were infected with serial tenfold dilution of supernatant media from infected cells and inhibitor. Plates were incubated for 1 h at 37 °C to allow viral adsorption. Then, 2 mL/well of overlay [MEM containing 2% FBS and 0.4% tragacanth (Merck, Jerusalem, Israel)] was added to each well and plates were incubated at 37 °C, 5% CO_2_ for 72 h. The media were then aspirated, and the cells were fixed and stained with 1 mL/well of crystal violet solution (Biological Industries, Israel). The inhibitory capacity of DTBN (1 mM) was assessed by determining the numbers of plaques compared with untreated cells. All experiments involving SARS-CoV-2 were conducted in a BSL3 facility in accordance with the Israel Institute for Biological Research regulation.

### 4.4. Animal Experiments

All animal experiments involving SARS-CoV-2 were conducted in a biosafety level 3 (BSL3) facility. The animal handling was performed in accordance with the regulations outlined in the U.S. Department of Agriculture (USDA) Animal Welfare Act and the conditions specified in the Guide for Care and Use of Laboratory Animals (National Institutes of Health, 2011). The animal studies were approved by the ethical committee for animal experiments of the Israel Institute for Biological Research (IIBR) (protocol number M-77-20). Female K18-hACE2 transgenic mice (B6. Cg-Tg(K18-ACE2)2Prlmn/J; #034860) (Jackson Laboratory, Bar Harbor, USA) (6–8 weeks old) were maintained at 20–22 °C and a relative humidity of 50 ± 10% under a 12 h light/dark cycle. The animals were fed commercial rodent chow (Koffolk Inc. Ramat Hovav, Israel) and provided tap water ad libitum. Prior to infection, the mice were kept in groups of 10. The mice were randomly assigned to the experimental groups. For SARS-CoV-2 infection, the virus was diluted in phosphate-buffered saline (PBS) supplemented with 2% FBS (Biological Industries, Israel). Anesthetized animals (ketamine 75 mg/kg and xylazine 7.5 mg/kg in PBS) were infected by intranasal (i.n.) instillation of 20 µL SARS-CoV-2 500 PFU/mouse and injected i.p. with DTBN, 20 μg/mouse once every day for 5 days, in 0.1 mL of 10% saline: DMSO (vehicle). 

### 4.5. Immunochemistry and Evaluation of Viral Presence in the Lung Sections

Lungs obtained 3 days after viral infection and concomitant daily treatment with DTBN or vehicle were immersed overnight in 10% neutral buffered formalin, trimmed and processed according to standard protocols [36]. Histology sections from paraffin blocks were cut at 5–6 μm on a rotary microtome, mounted onto glass slides, and stained with H&E. Tissue examination was performed by an experienced pathologist (DB). 

For immunofluorescence, lung sections of the same mice were stained with a rabbit polyclonal antibody developed in our laboratory against SARS-CoV-2, rabbits were inoculated five times intravenously with 10^6^ PFU of live SARS-CoV-2 at days 0, 7, 10, 14, and 17. Serum was collected 14 days after the final dose. To visualize the virus, a secondary red fluorescent antibody anti-Rabbit IgG (Jackson Immuno Research Inc., Bar Harbor, USA) was used. 

Red fluorescence in the background of DAPI stained sections (blue) was quantified as follows:

In order to segment the scanned images and classify the resulting objects, the semantic segmentation and classification of Ilastik [37] was used. As a result, highly accurate masks for both the tissue and the virus particles were produced. Then, the masked images were analyzed in Fiji (‘Analyse particles’) [38] to retrieve the quantities of interest of each object (area, intensity, principal axes, etc.). After that, the ratio of total tissue area to total virus area was calculated for each image. The lungs of 4 mice and 3 lungs from DTBN-treated mice were scanned and analyzed independently (Appendix A). The unpaired *t*-test of the area ratios was computed, showing a significant difference between the control and the two treated groups (*p* value = 0.0215). 

### 4.6. Computational Methodology

#### 4.6.1. The Molecular Docking and MM-GBSA Refinement

The molecular docking studies were performed using Schrodinger Maestro 2021-2 (Schrödinger, LLC, New York, NY, USA). DTBN was docked to the catalytic site of RdRp using the known inhibitor suramin-bound RdRp 3-D structure as a reference (PDB: 7D4F). Both protein and ligand structures were optimized before docking. Inconsistencies in the structure were corrected during the optimization process. The protein preparation wizard and LigPrep module of Schrodinger Maestro 2021-2 (Schrödinger, LLC, New York, NY, USA) were used for the optimization process. Post-docking MM-GBSA energies calculation was carried out on the docked poses with a flexible residue distance of 5 Å.

The binding energy was calculated by the following formula:ΔG = Σ(ΔG_Bind coulomb_ + ΔG_Bind covalent_ + ΔG_Bind H-bond_ + ΔG_Bind Lipo_ + ΔG_Bindpacking_ + ΔG_Bind Self cont_ + ΔG_Bind Solv GB_ + ΔG_Bind vdw_)(1)

#### 4.6.2. Molecular Dynamics (MD) Simulation

MD simulation for a duration of 100 ns was performed using a DESMOND module by Schrodinger (Schrodinger release 2021). Through the system builder panel, DTBN and suramin with an RdRp protein complex were immersed in an orthorhombic box of the SPC solvent model. The solvated system was neutralized using counter ions and a physiological NaCl salt concentration of 0.15 M. OPLS4 force field was utilized. The simulation was 100 ns using NPT assemble class at a temperature of 300 K and an atmospheric pressure of 1.013 bar.

## Figures and Tables

**Figure 1 ijms-24-08327-f001:**
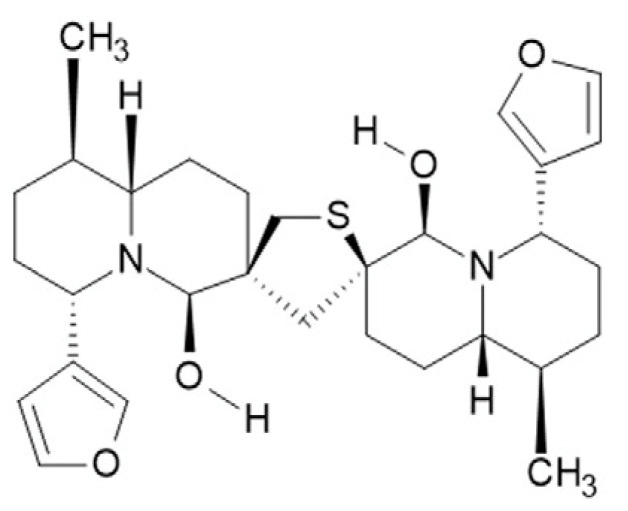
Structure of 6,6′-dihydroxythiobinupharidine (DTBN).

**Figure 2 ijms-24-08327-f002:**
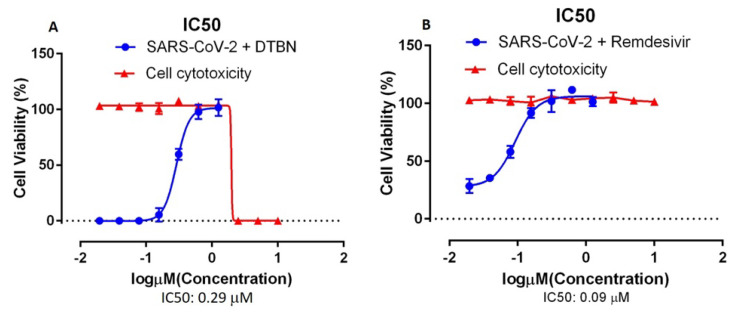
Half maximal inhibitory concentration (IC50) of 6,6-Dihydroxythiobinupharidine (DTBN) against SARS-CoV-2 infection. (**A**) Vero E6 cells were incubated with DTBN for 1 h and then infected with SARS-CoV-2 at an MOI of 0.01 in the presence of the indicated drug concentration (0.15−10 μM) for 72 h. IC50-0.29 μM (blue line) was determined from dose response curve based on treatment with eight concentrations based on cell viability (XTT) assay. Vero E6 cells were incubated with DTBN for 72 h in the presence of the indicated drug concentration (10 μM−0.15 μM) and cell viability (red line) was determined. (**B**) A similar positive control was performed on the same cells with remdesivir IC50: 0.09 μM.

**Figure 3 ijms-24-08327-f003:**
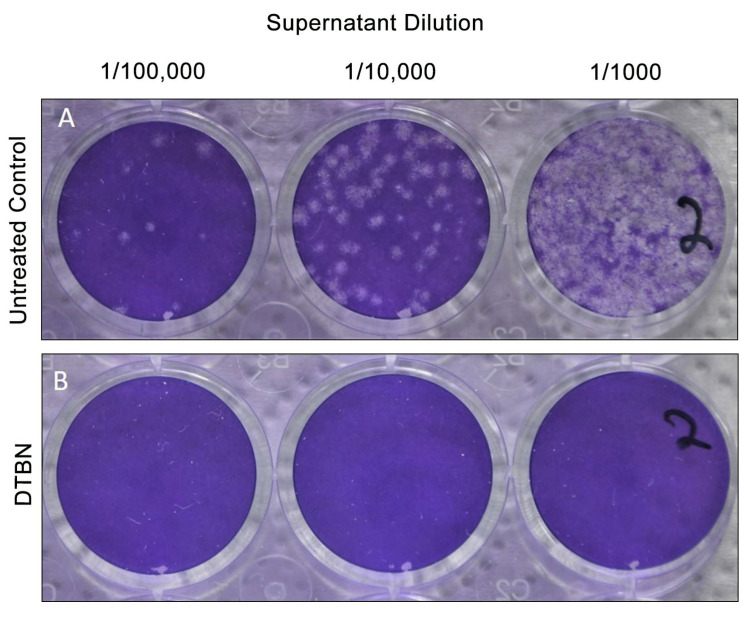
Plaque forming unit 1 h pre-infection. Vero E6 cells were treated with DTBN (1 mM) and were infected 1 h later with SARS-CoV-2 (MOI, 0.01). The supernatant was collected 24 h after infection for PFU quantification. (**A**) Control: supernatants of infected cells treated with vehicle, DMSO. (**B**) Supernatant of infected cells treated with DTBN (1 mM).

**Figure 4 ijms-24-08327-f004:**
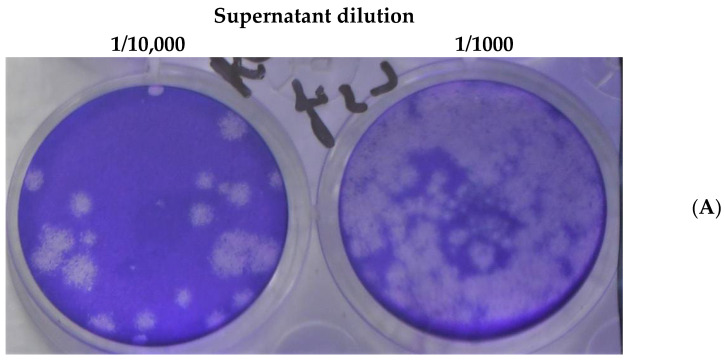
Plaque-forming units 24 h post-infection. Vero E6 cells were infected with SARS-CoV-2 (MOI, 0.01) and, 24 h later, treated with DTBN (1 mM). The supernatant was collected 48 h after infection for PFU quantification. (**A**) Control: supernatant of infected cells treated with vehicle, DMSO. (**B**) Supernatant of infected cells treated with DTBN (1 mM).

**Figure 5 ijms-24-08327-f005:**
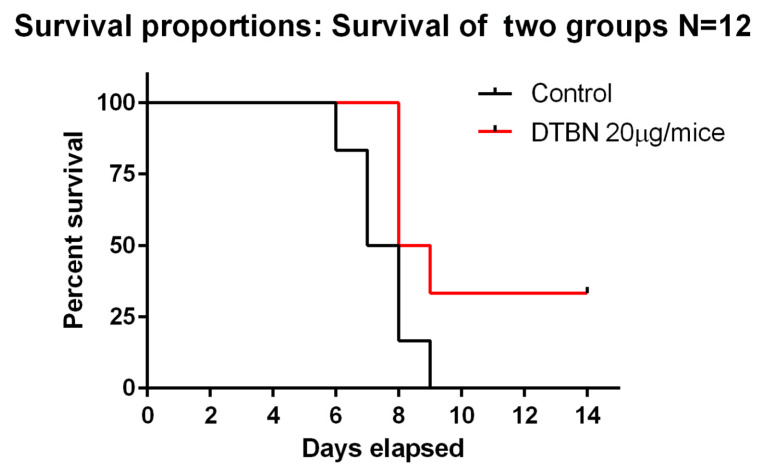
In-vivo assay. Anesthetized animals were infected by i.n. instillation of 20 µL SARS-CoV-2 500 PFU/mouse and injected i.p. with DTBN, 20 μg/mouse in 10% saline: DMSO once a day for 5 days (n = 12). Red line—treated group; black line—vehicle treated control (n = 12). Log-rank *p* = 0.047.

**Figure 6 ijms-24-08327-f006:**
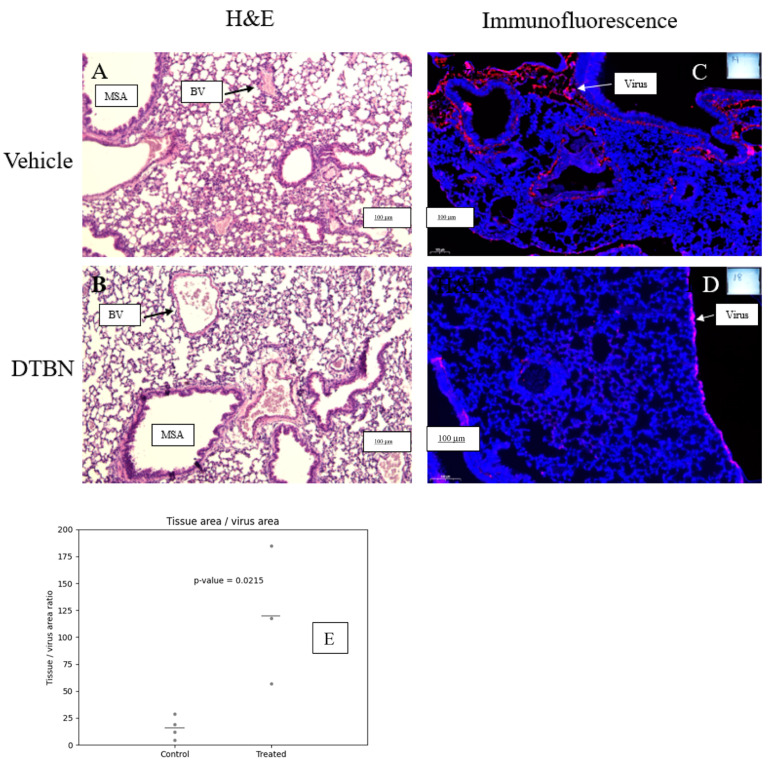
Expression of viral particles in lungs of vehicle- and DTBN-treated mice. Lungs from vehicle- or DTBN-treated mice were prepared for H&E and immunofluorescence. H&E staining of lung sections from vehicle- (**A**) and DTBN- (**B**) treated mice. Immunofluorescence. The sections were stained first with DAPI (blue) and then with a polyclonal antibody against the virus and a secondary anti-Ig antibody (red). Representative sections from vehicle- (**C**) and DTBN- (**D**) treated mice are shown. (**E**) Quantification of the tissue (blue)/virus (pink) area ratio. T-test analysis. Medium size airway (MSA), blood vessel (BV). 100 μm bars are noted.

**Figure 7 ijms-24-08327-f007:**
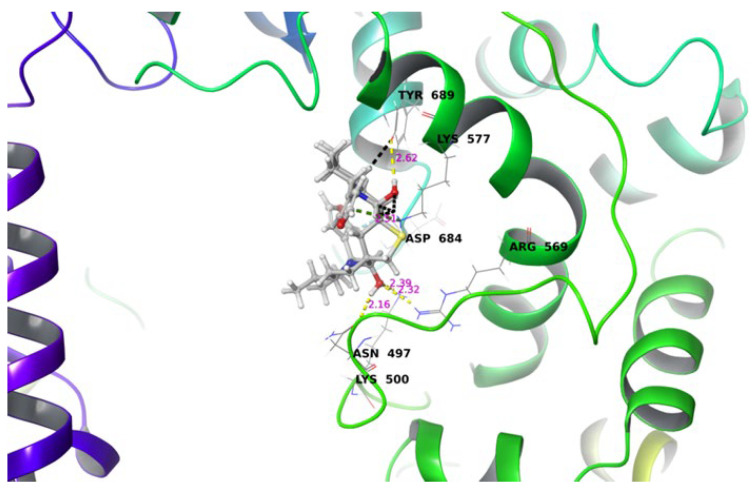
DTBN bound to the catalytic site of RdRp.

**Figure 8 ijms-24-08327-f008:**
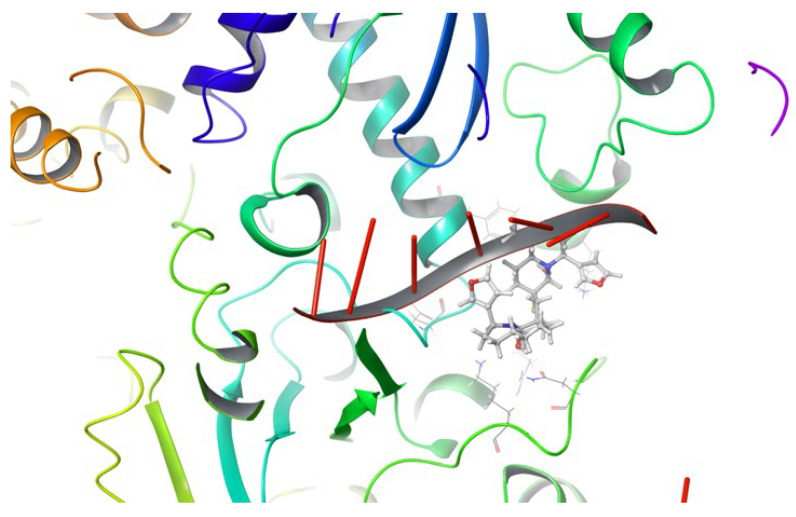
DTBN and RNA Template overlapped at the catalytic site of RdRp.

**Figure 9 ijms-24-08327-f009:**
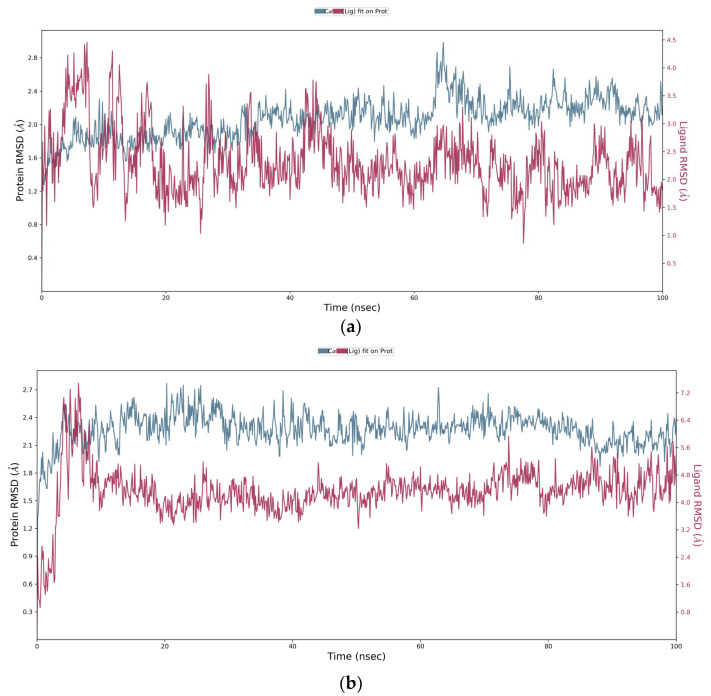
(**a**) RMSD graph of DTBN–RdRp complex. (**b**) RMSD graph of suramin–RdRp complex.

**Figure 10 ijms-24-08327-f010:**
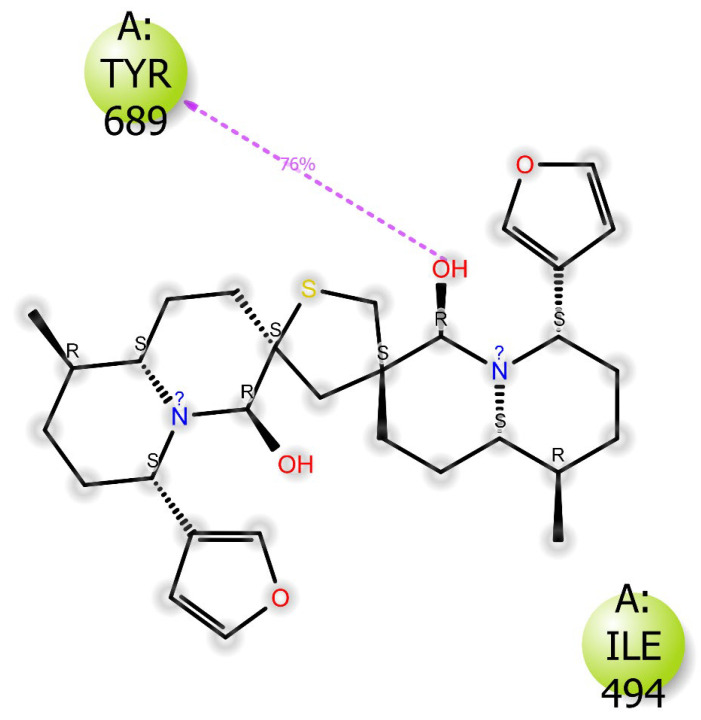
DTBN–RdRp protein interactions with >50% strength.

**Figure 11 ijms-24-08327-f011:**
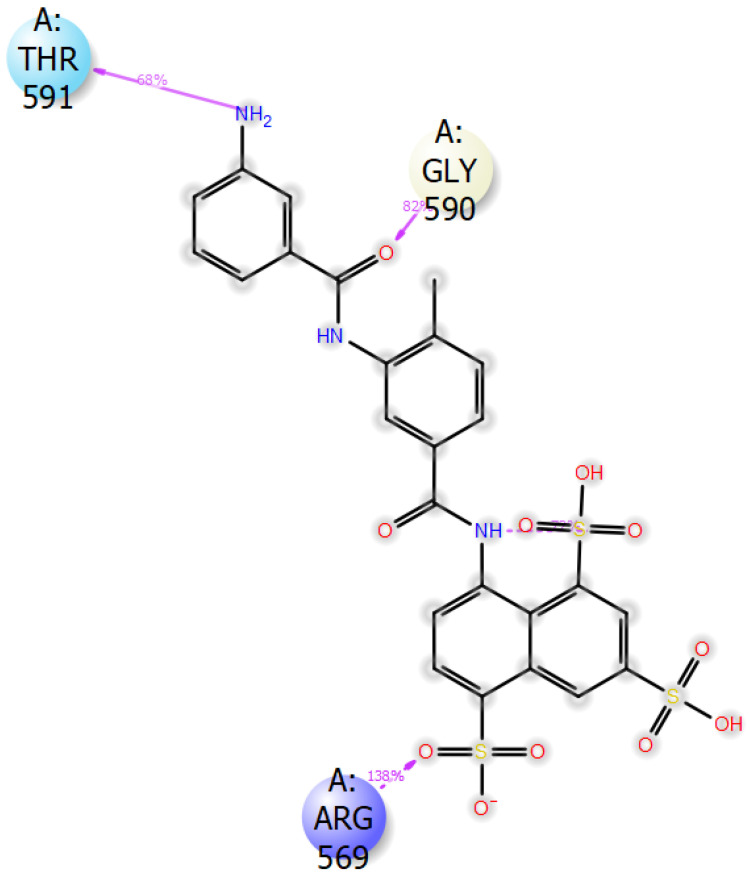
Suramin–RdRp protein interactions with >50% strength.

**Table 1 ijms-24-08327-t001:** RdRp domains.

Motif	Range
Motif G	499–511
Motif F	544–560
Motif A	612–626
Motif B	678–710
Motif C	753–767
Motif D	771–796
Motif E	810–820

**Table 2 ijms-24-08327-t002:** In Silico Molecular Docking.

S.No.	Compound	RdRp ΔG (Kcal/mol)	ACE2 ΔG (Kcal/mol)
1.	DTBN	−38.26	−27.88
2.	Control	Suramin (−59.15)	MLN-4760 (−73.89)

## Data Availability

All data is available from the corresponding author (J.G.) on request.

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
