# Peer review of "In Vitro and In Vivo Therapeutic Potential of 6,6′-Dihydroxythiobinupharidine (DTBN) from Nuphar lutea on Cells and K18-hACE2 Mice Infected with SARS-CoV-2"

_ijms, 2023, doi:10.3390/ijms24098327_

Round 1
Reviewer 1 Report
Comments
The article entitled “In vitro and in vivo therapeutic potential of 6,6′-dihydroxythi-obinupharidine (DTBN) from Nuphar lutea on cells and K18- hACE mice infected with SARS-CoV-2” the authors have showed the promising activity for novel molecule DTBN through preclinical study in vitro and in vivo should against the pathogens of SARS-CoV-2.
However, there are some minor corrections in this manuscript which are needed.
1. Authors have performed Molecular docking studies, since these studies are performed with less/no flexibility in the protein or ligand securing an best docked pose is difficult. It would have been nice if they could reproduce the docked pose relevant to any crystal structure ligand molecule for confirmation.
2. Authors have mentioned in the methods of Molecular dynamics they have performed 100ns production run. The results highlighted shows only 20ns, this is very insufficient time to even understand the evolution of the dynamics. And performing one run is always ruled out due to the stochastic nature of MD simulation, sampling very less dynamics in a conformational space.
3. In line with the above question, the RMSD values aren't converged at a short time scale. There is lot of dynamics in the protein suggested that its needs to converged. This would be achieved only when the simulations are extended.
Decision: Accept after mentioned minor revision.

Author Response
As requested, the wording of certain paragraphs marked in yellow was changed. The only paragraph that was not changed is 2.4. Animal experiments. The wording of the animal experimental protocol has not been changed since it is purely technical with almost no variation from other work by us.
Reviewer 1
The manuscript Weiss et al. describes antiviral properties of 6,6′-dihydroxythiobinupharidine DTBN, isolated from the Nuphar lutea extract, on Covid-19. In vitro and in vivo experiments were conducted. DTNB effectively inhibits SARS-CoV-2 production in Vero E6 cells and lowered virus levels in the lungs and improved survival of infected mice. The potential target was proposed. Molecular docking and molecular dynamic simulation data were given.
The obtained results are of scientific interest and the manuscript can be published in IJMS after revision.
Authors need to add a table with data on activity and cytotoxicity of DTBN on Vero E6 cells as well as activity of Remdesivir in the same conditions or other known control. I didn’t find information which control was used in your experiments.
Cytotoxicity of DTBN on Vero E6 cells is presented in figure 2. A positive control of the same cells in the same conditions with Remedesivir was added as figure 2B.
Authors proposed several targets including RdRp. Molecular docking to SARS-COV-2 RdRp was conducted. It would be better to see the inhibitory activity on the isolated RdRp or at least against other SARS-COV-2 targets.
Indeed, validation of the molecular docking analysis by demonstrating inhibition of isolated enzymes by DTBN is an important step in determining a target molecule. In this manuscript we demonstrated the anti- SARS-COV-2 activity of DTBN as a proof-of-concept work and suggested possible targets. Further work demonstrating which are these targets and the mechanism of action is necessary but not in the scope of this paper.
Paragraph from line 417 to line 422 needs revision (typos).
The paragraph was revised and corrected
However, there are some minor corrections in this manuscript which are needed.
- Authors have performed Molecular docking studies, since these studies are performed with less/no flexibility in the protein or ligand securing an best docked pose is difficult. It would have been nice if they could reproduce the docked pose relevant to any crystal structure ligand molecule for confirmation.
As per reviewer’s suggestion, the docked pose for crystal structures for suramin-RdRp crystal structures is reproduced and results are incorporated in the form of text in the manuscript “ To validate the docking protocol……..”. 3D image of superimposition and 2D interaction of Crystal Ligand and reproduced docked Pose is given in supplementary Figure 2 and Figure 3.
- Authors have mentioned in the methods of Molecular dynamics they have performed 100ns production run. The results highlighted shows only 20ns, this is very insufficient time to even understand the evolution of the dynamics. And performing one run is always ruled out due to the stochastic nature of MD simulation, sampling very less dynamics in a conformational space.
The authors thank the reviewer for the critical review, the correction has been made and results for 100ns have been incorporated in the manuscript. Figure 9a 9b, 10 and 11 has also been revised accordingly. The authors agree with the reviewer’s suggestion but due to the large size of the Protein-ligand complex system (>120000 atoms) and limited computational power resource availability, we currently can perform only a single run. Therefore, we request the reviewer to kindly consider the limitation.
- In line with the above question, the RMSD values aren't converged at a short time scale. There is lot of dynamics in the protein suggested that its needs to converged. This would be achieved only when the simulations are extended.
As per the reviewer’s suggestion the time scale of the dynamic simulation have been extended from 20ns to 100ns and results for the same have been incorporated in the manuscript.
Reviewer 2 Report
The manuscript Weiss et al. describes antiviral properties of 6,6′-dihydroxythiobinupharidine DTBN, isolated from the Nuphar lutea extract, on Covid-19. In vitro and in vivo experiments were conducted. DTNB effectively inhibits SARS-CoV-2 production in Vero E6 cells and lowered virus levels in the lungs and improved survival of infected mice. The potential target was proposed. Molecular docking and molecular dynamic simulation data were given.
The obtained results are of scientific interest and the manuscript can be published in IJMS after revision.
Authors need to add a table with data on activity and cytotoxicity of DTBN on Vero E6 cells as well as activity of Remdesivir in the same conditions or other known control. I didn’t find information which control was used in your experiments.
Authors proposed several targets including RdRp. Molecular docking to SARS-COV-2 RdRp was conducted. It would be better to see the inhibitory activity on the isolated RdRp or at least against other SARS-COV-2 targets.
Paragraph from line 417 to line 422 needs revision (typos).
Author Response
Major
- Figure 3. There is no correspondence between the figure 3 and the main text. Indeed, in the main text (lines 230-232) the authors say: “Approximately 7x106 PFU/ml were detected in the medium of vehicle di-methyl sulfoxide (DMSO) (untreated) infected cells, whereas no plaques were detected in DTBN treated cells, indicating significant inhibition of virus release”. However, figure 3 does not show this difference in 1h pre-infection between vehicle and DTBN treated cells, but only demonstrates the plaque forming at different dilutions.
Moreover, in figure legend letter A and B are indicated, which are absent in figure I guess, the figure 3 is wrong. Please, discuss this point.
Corrected, A and B were added.
- Paragraph 3.3. the authors write: “Approximately 3E5 PFU/ml were detected in the medium of vehicle 241 dimethyl sulfoxide (DMSO) (untreated) infected cells, whereas no plaques were detected 242 in DTBN treated cells”. Did the authors refer to figure 4? It is not indicated which figure main text is referring to. Again, here, I don’t get the meaning of Figure 3, which is still indicated in this paragraph.
Corrected
- Line 74-85. When discussing about the SARS-CoV-2 as responsible for COVID-19, I would suggest to mention (among the other characteristics of the virus) the bacteriophage behavior of SARS-CoV-2, as very recently demonstrated by the authors of this study (doi.org/10.3390/ijms24043929). I recommend to cite this article in order to make the present article more updated.
The paper was added
- Figure 6 A and B. Maybe it would help the readers if the authors could add an example of H&E of control uninfected mice in order to understand how the normal lung histology should appear without viral infection. This further image could be added as supplementary image.
The image was added as figure 1 in the supplementary materials section
- Figure 6 C and D: immunofluorescence images. It is not specified which protein of SARS-CoV-2 is recognized by the primary antibody used. Please, clarify this part.
As stated in: 2.5. Immunohistochemistry and evaluation of viral presence in the lung sections. The primary antibody is a rabbit polyclonal antibody against the whole virus, therefore it is assummed that it binds several viral antigens.
Regarding the quantification of tissue/virus ratio it would be important to add in the main article a graph of the quantification showing histograms and the statistical analysis.
The tissue virus ratio quantification was added as figure 6E
Moreover, in supplementary table, it is not indicated which are vehicle and DTBN treated mice.
The vehicle and DTBN labeling in the supplementary table were added
In addition, all the scale bars of figure 6 are not visible: please, re-add them and mention also in figure legend.
The bars were added and mentioned in the figure legend
- For in vivo experiments, female mice were used. Please, discuss why female but not male animals were used.
A similar in vivo experiment was performed with no difference in the results between male and female mice: https://www.ncbi.nlm.nih.gov/pmc/articles/PMC7878817/
Therefore, female mice were used.
- Figure 10 and figure 11 are not indicated in the main text when the authors discuss about the strength of the interaction in RdRp-DTBN and RdRp-Surammin complexes
Figure 10 and figure 11 were added to the text
Minor:
- Line 71-73. “Several recent thorough reviews have been published [3–5] on the state-of-the-art status of small and peptide-based molecules as well as therapeutic monoclonal antibodies against COVID-19.
I would suggest to add a very recent review regarding the development of novel molecules to be used as therapeutic options against bacteria and viruses.
The following 2023 paper was added.
Shoaib Shoaib,1 Mohammad Azam Ansari,2,* Geetha Kandasamy,3 Rajalakshimi Vasudevan,4 Umme Hani,5 Waseem Chauhan,6 Maryam S. Alhumaidi,7 Khadijah A. Altammar,7 Sarfuddin Azmi,8 Wasim Ahmad,9 Shadma Wahab,10 and Najmul Islam1,* An Attention towards the Prophylactic and Therapeutic Options of Phytochemicals for SARS-CoV-2: A Molecular Insight. Molecules. 2023 Jan; 28(2): 795.
- Throughout the manuscript, Covid-19 is reported both with lower case and upper case (COVID-19). Please, uniform this discrepancy by inserting only COVID-19. In addition, COVID-19 should be written with full name at the first mention.
Corrected
- Line 98. “The Food and Drug Administration approved remdesivir, an inhibitor of
RdRp, for the treatment of COVID-19 and issued an emergency use authorization for molnupiravir also an RdRp inhibitor as well as nirmatrelvir an inhibitor of the main protease.
I recommend the authors to cite these references regarding their statement.
https://www.fda.gov/news-events/press-announcements/fda-approves-first-treatment-covid-19 (for remdesivir)
https://www.fda.gov/news-events/press-announcements/coronavirus-covid-19-update-fda-authorizes-additional-oral-antiviral-treatment-covid-19-certain (for molnupiravir)
https://www.fda.gov/news-events/press-announcements/coronavirus-covid-19-update-fda-authorizes-first-oral-antiviral-treatment-covid-19 (for nirmatrelvir)
The references were added
- Line 167: substitute “DTNB” with “DTBN”
Corrected
- Figure 5: in the legend is indicated a concentration of DTBN of 20ug/mice, while in other cases is reported 20 mg/mice. Please, correct the mistakes.
Corrected to 20ug/mice
- Line 289-290: The term “SARS-COVID-2” is not correct to me: indeed, if the authors are referring to the virus, it would be more appropriate to use only the term “SARS-CoV-2”. Please, check and correct this mistake throughout all the manuscript.
Corrected to SARS-CoV-2
- The terms “in vitro” and “in vivo” should always be reported in italics. Please, check and correct throughout the manuscript
Corrected
- Line 328. Remove full stop after DTBN
Corrected
- The numbers of paragraphs 3.6 and 3.7 are in bold font. Please, uniform these numbers with all other paragraphs.
Corrected
Reviewer 3 Report
In this study the authors tested the therapeutic ability of DTBN, a molecule isolated from Nuphar lutea extract, to reduce the levels of SARS-CoV-2 on in vitro and in vivo models. DTNB was able to inhibit viral production on Vero cells treated 1h before infection and 24 h post infection. Mice infected with virus also showed a delayed onset of clinical outcome, as confirmed by histological sections. The work is interesting and novel. The English is very good. However, I have many major concerns that should be carefully addressed by the authors before I consider their article acceptable for a publication.
Major
· Figure 3. There is no correspondence between the figure 3 and the main text. Indeed, in the main text (lines 230-232) the authors say: “Approximately 7x106 PFU/ml were detected in the medium of vehicle di-methyl sulfoxide (DMSO) (untreated) infected cells, whereas no plaques were detected in DTBN treated cells, indicating significant inhibition of virus release”. However, figure 3 does not show this difference in 1h pre-infection between vehicle and DTBN treated cells, but only demonstrats the plaque forming at different dilutions. Moreover, in figure legend letter A and B are indicated, which are absent in figure. I guess, the figure 3 is wrong. Please, discuss this point.
· Paragraph 3.3. the authors write: “Approximately 3E5 PFU/ml were detected in the medium of vehicle 241 dimethyl sulfoxide (DMSO) (untreated) infected cells, whereas no plaques were detected 242 in DTBN treated cells”. Did the authors refer to figure 4? It is not indicated which figure main text is referring to. Again, here, I don’t get the meaning of Figure 3, which is still indicated in this paragraph.
· Line 74-85. When discussing about the SARS-CoV-2 as responsible for COVID-19, I would suggest to mention (among the other characteristics of the virus) the bacteriophage behavior of SARS-CoV-2, as very recently demonstrated by the authors of this study (doi.org/10.3390/ijms24043929). I recommend to cite this article in order to make the present article more updated.
· Figure 6 A and B. Maybe it would help the readers if the authors could add an example of H&E of control uninfected mice in order to understand how the normal lung histology should appear without viral infection. This further image could be added as supplementary image.
· Figure 6 C and D: immunofluorescence images. It is not specified which protein of SARS-CoV-2 is recognized by the primary antibody used. Please, clarify this part. Regarding the quantification of tissue/virus ratio it would be important to add in the main article a graph of the quantification showing histograms and the statistical analysis. Moreover, in supplementary table, it is not indicated which are vehicle and DTBN treated mice.
In addition, all the scale bars of figure 6 are not visible: please, re-add them and mention also in figure legend.
· For in vivo experiments, female mice were used. Please, discuss why female but not male animals were used.
· Figure 10 and figure 11 are not indicated in the main text when the authors discuss about the strength of the interaction in RdRp-DTBN and RdRp-Surammin complexes
Minor:
· Line 71-73. “Several recent thorough reviews have been published [3–5]on the state-of-the-art status of small and peptide-based molecules as well as therapeutic monoclonal antibodies against COVID-19.
I would suggest to add a very recent review regarding the development of novel molecules to be used as therapeutic options against bacteria and viruses.
· Throughout the manuscript, Covid-19 is reported both with lower case and upper case (COVID-19). Please, uniform this discrepancy by inserting only COVID-19. In addition, COVID-19 should be written with full name at the first mention.
· Line 98. “The Food and Drug Administration approved remdesivir, an inhibitor of
RdRp, for the treatment of COVID-19 and issued an emergency use authorization for molnupiravir also an RdRp inhibitor as well as nirmatrelvir an inhibitor of the main protease.
I recommend the authors to cite these references regarding their statement.
https://www.fda.gov/news-events/press-announcements/fda-approves-first-treatment-covid-19 (for remdesivir)
https://www.fda.gov/news-events/press-announcements/coronavirus-covid-19-update-fda-authorizes-additional-oral-antiviral-treatment-covid-19-certain (for molnupiravir)
https://www.fda.gov/news-events/press-announcements/coronavirus-covid-19-update-fda-authorizes-first-oral-antiviral-treatment-covid-19 (for nirmatrelvir)
· Line 167: substitute “DTNB” with “DTBN”
· Figure 5: in the legend is indicated a concentration of DTBN of 20ug/mice, while in other cases is reported 20 mg/mice. Please, correct the mistakes.
· Line 289-290: The term “SARS-COVID-2” is not correct to me: indeed, if the authors are referring to the virus, it would be more appropriate to use only the term “SARS-CoV-2”. Please, check and correct this mistake throughout all the manuscript.
· The terms “in vitro” and “in vivo” should always be reported in italics. Please, check and correct throughout the manuscript
· Line 328. Remove full stop after DTBN
· The numbers of paragraphs 3.6 and 3.7 are in bold font. Please, uniform these numbers with all other paragraphs.
Author Response
Major
Figure 3. There is no correspondence between the figure 3 and the main text. Indeed, in the main text (lines 230-232) the authors say: “Approximately 7x106 PFU/ml were detected in the medium of vehicle di-methyl sulfoxide (DMSO) (untreated) infected cells, whereas no plaques were detected in DTBN treated cells, indicating significant inhibition of virus release”. However, figure 3 does not show this difference in 1h pre-infection between vehicle and DTBN treated cells, but only demonstrates the plaque forming at different dilutions.
Moreover, in figure legend letter A and B are indicated, which are absent in figure I guess, the figure 3 is wrong. Please, discuss this point.
Corrected, A and B were added.
- Paragraph 3.3. the authors write: “Approximately 3E5 PFU/ml were detected in the medium of vehicle 241 dimethyl sulfoxide (DMSO) (untreated) infected cells, whereas no plaques were detected 242 in DTBN treated cells”. Did the authors refer to figure 4? It is not indicated which figure main text is referring to. Again, here, I don’t get the meaning of Figure 3, which is still indicated in this paragraph.
Corrected
- Line 74-85. When discussing about the SARS-CoV-2 as responsible for COVID-19, I would suggest to mention (among the other characteristics of the virus) the bacteriophage behavior of SARS-CoV-2, as very recently demonstrated by the authors of this study (doi.org/10.3390/ijms24043929). I recommend to cite this article in order to make the present article more updated.
The paper was added
- Figure 6 A and B. Maybe it would help the readers if the authors could add an example of H&E of control uninfected mice in order to understand how the normal lung histology should appear without viral infection. This further image could be added as supplementary image.
The image was added as figure 1 in the supplementary materials section
- Figure 6 C and D: immunofluorescence images. It is not specified which protein of SARS-CoV-2 is recognized by the primary antibody used. Please, clarify this part.
As stated in: 2.5. Immunohistochemistry and evaluation of viral presence in the lung sections. The primary antibody is a rabbit polyclonal antibody against the whole virus, therefore it is assummed that it binds several viral antigens.
Regarding the quantification of tissue/virus ratio it would be important to add in the main article a graph of the quantification showing histograms and the statistical analysis.
The tissue virus ratio quantification was added as figure 6E
Moreover, in supplementary table, it is not indicated which are vehicle and DTBN treated mice.
The vehicle and DTBN labeling in the supplementary table were added
In addition, all the scale bars of figure 6 are not visible: please, re-add them and mention also in figure legend.
The bars were added and mentioned in the figure legend
- For in vivo experiments, female mice were used. Please, discuss why female but not male animals were used.
A similar in vivo experiment was performed with no difference in the results between male and female mice: https://www.ncbi.nlm.nih.gov/pmc/articles/PMC7878817/
Therefore, female mice were used.
- Figure 10 and figure 11 are not indicated in the main text when the authors discuss about the strength of the interaction in RdRp-DTBN and RdRp-Surammin complexes
Figure 10 and figure 11 were added to the text
Minor:
- Line 71-73. “Several recent thorough reviews have been published [3–5] on the state-of-the-art status of small and peptide-based molecules as well as therapeutic monoclonal antibodies against COVID-19.
I would suggest to add a very recent review regarding the development of novel molecules to be used as therapeutic options against bacteria and viruses.
The following 2023 paper was added.
Shoaib Shoaib,1 Mohammad Azam Ansari,2,* Geetha Kandasamy,3 Rajalakshimi Vasudevan,4 Umme Hani,5 Waseem Chauhan,6 Maryam S. Alhumaidi,7 Khadijah A. Altammar,7 Sarfuddin Azmi,8 Wasim Ahmad,9 Shadma Wahab,10 and Najmul Islam1,* An Attention towards the Prophylactic and Therapeutic Options of Phytochemicals for SARS-CoV-2: A Molecular Insight. Molecules. 2023 Jan; 28(2): 795.
- Throughout the manuscript, Covid-19 is reported both with lower case and upper case (COVID-19). Please, uniform this discrepancy by inserting only COVID-19. In addition, COVID-19 should be written with full name at the first mention.
Corrected
- Line 98. “The Food and Drug Administration approved remdesivir, an inhibitor of
RdRp, for the treatment of COVID-19 and issued an emergency use authorization for molnupiravir also an RdRp inhibitor as well as nirmatrelvir an inhibitor of the main protease.
I recommend the authors to cite these references regarding their statement.
https://www.fda.gov/news-events/press-announcements/fda-approves-first-treatment-covid-19 (for remdesivir)
https://www.fda.gov/news-events/press-announcements/coronavirus-covid-19-update-fda-authorizes-additional-oral-antiviral-treatment-covid-19-certain (for molnupiravir)
https://www.fda.gov/news-events/press-announcements/coronavirus-covid-19-update-fda-authorizes-first-oral-antiviral-treatment-covid-19 (for nirmatrelvir)
The references were added
- Line 167: substitute “DTNB” with “DTBN”
Corrected
- Figure 5: in the legend is indicated a concentration of DTBN of 20ug/mice, while in other cases is reported 20 mg/mice. Please, correct the mistakes.
Corrected to 20ug/mice
- Line 289-290: The term “SARS-COVID-2” is not correct to me: indeed, if the authors are referring to the virus, it would be more appropriate to use only the term “SARS-CoV-2”. Please, check and correct this mistake throughout all the manuscript.
Corrected to SARS-CoV-2
- The terms “in vitro” and “in vivo” should always be reported in italics. Please, check and correct throughout the manuscript
Corrected
- Line 328. Remove full stop after DTBN
Corrected
- The numbers of paragraphs 3.6 and 3.7 are in bold font. Please, uniform these numbers with all other paragraphs.
Corrected
Round 2
Reviewer 2 Report
The authors added data from experiments with control. The article now may be published in IJMS.
Reviewer 3 Report
The authors have addressed all my concerns, so I suggest this article for publication. Thank you!